# Transcription Factors as Targets of Natural Compounds in Age-Related Diseases and Cancer: Potential Therapeutic Applications

**DOI:** 10.3390/ijms232213882

**Published:** 2022-11-10

**Authors:** Mi Eun Kim, Dae Hyun Kim, Jun Sik Lee

**Affiliations:** 1Department of Life Science, Immunology Research Lab, BK21-plus Research Team for Bioactive Control Technology, College of Natural Sciences, Chosun University, Dong-gu, Gwangju 61452, Korea; 2LKBio Inc., Chosun University Business Incubator (CUBI) Building, Dong-gu, Gwangju 61452, Korea

**Keywords:** inflammation, natural compounds, age, cancer, transcription factors

## Abstract

Inflammation exacerbates systemic pathophysiological conditions and chronic inflammation is a sustained and systemic phenomenon that aggravates aging that can lead to chronic age-related diseases. These inflammatory phenomena have recently been redefined and delineated at the molecular, cellular, and systemic levels. Many transcription factors that are activated in response to tumor metabolic state have been reported to be regulated by a class of histone deacetylase called sirtuins (SIRTs). Sirtuins play a pivotal role in the regulation of tumor cell metabolism, proliferation, and angiogenesis, including oxidative stress and inflammation. The SIRT1-mediated signaling pathway in diabetes and cancer is the SIRT1/forkhead-box class O (FoxO)/nuclear factor-kappa B (NF-κB) pathway. In this review, we describe the accumulation of SIRT1-, NF-κB-, and FoxO-mediated inflammatory processes and cellular proinflammatory signaling pathways. We also describe the proinflammatory mechanisms underlying metabolic molecular pathways in various diseases such as liver cancer and diabetes. Finally, the regulation of cancer and diabetes through the anti-inflammatory effects of natural compounds is highlighted. Evidence from inflammation studies strongly suggests that cells may be a major source of cytokines secreted during various diseases. A better understanding of the mechanisms that underpin the inflammatory response and palliative role of natural compounds will provide insights into the molecular mechanisms of inflammation and various diseases for potential intervention.

## 1. Introduction

Inflammation results in increased circulating levels of cytokines and chemokines, including TNF-α, IL-1β, IL-6, MCP-1, MIP-1α, RANTES, and IL-18 in monocytes, macrophages, and microphages, and increased expression of genes involved in inflammation [1,2]. Various cytokines can cause chronic inflammation [2]. Inflammation causes collateral damage to tissues and organs over time by inducing oxidative stress [3,4]. Acute inflammatory responses are also activated by damage-associated molecular patterns secreted in the presence of extracellular stress, including chemical or metabolically injurious stimuli under cellular stress or damage [5]. However, the clinical consequences of inflammation damage can be severe, and there is a risk of metabolic syndrome with hyperglycemia, type 2 diabetes [6], and various types of cancer [7]. Experimental data strongly suggest that inflammation plays a critical role in the onset and progression of metabolic syndrome. Inflammation associated with diabetes and liver cancer has been found to share similar risk factors, including metabolic abnormalities. Inflammatory molecules such as TNF, IL-1β, IL-6, MCP-1, and IL-18 induces age-related chronic inflammation, leading to aging and cancers.

The aging process can be defined as progressive, physiological functional deterioration throughout the lifetime of an individual by different convoluted interactions among genes and non-genetic environmental factors that eventually result in disruption of homeostasis and increased susceptibility to disease or death. The basic mechanism of the aging process is a sustained, long-term inflammatory state that is further aggravated by elevated oxidative stress due to enhanced reactive oxygen species, lipid peroxidation, and protein oxidative modifications.

Sirtuin1 (SIRT1) regulates multiple physiological responses, including oxidative stress, apoptosis, and inflammation under various stress conditions [8]. Sirtuin1 regulates the activity of downstream targets including nuclear factor-kappa B (NF-κB) and forkhead-box class O (FoxO) proteins [8]. FoxO transcription factors influence and regulate cell cycle arrest, cell death, metabolism, DNA repair, oxidative stress resistance, and differentiation and may function as tumor suppressors [9]. FoxO proteins play a major role in the pathogenesis of both insulin resistance and cancer via insulin signaling [10].

Herbal medicines, derived from thousands of years of application and experience, have been used clinically in Eastern and Western countries. Herbal remedies are marketed and used as dietary supplements, but their quality and use are mostly unregulated and uncontrolled without scientific evidence of their mechanism of action and toxicity. Numerous natural products that possess antioxidant activities have been shown to display antioxidant properties such as the inhibition of lipid peroxidation, and DNA damage by activation of SIRT1. Consequently, modulation of sirtuin activity has been regarded as a promising therapeutic option for many pathologies [11].

In this review, the selection of experimentally tested herbs showing anti-inflammatory and antiaging effects in aging models, their active ingredients, and their targeting signaling pathways are reviewed together with available data and reports. This review summarizes the change effects of natural compounds on aging, inflammatory, and age-related diseases such as cancer.

## 2. Oxidative Stress, Inflammation, and Aging

Aging is a natural and inevitable part of life and is characterized by gradual declines in physiological functions that ultimately lead to morbidity and mortality. Among several well-known hypotheses of aging, the most accepted theory is that aging is caused by oxidative stress [12,13]. The oxidative stress hypothesis of aging describes the characteristic changes in the aging process as a net effect of redox imbalance caused by a difference between oxidative stress and a counter-acting, antioxidative force [14]. This redox imbalance is likely due to increases in reactive oxygen species (ROS) and reactive lipid aldehydes coupled with a weakened antioxidant defense system. Major culprits of the redox imbalance that occurs from age-related oxidative stress are elicited by the uncontrolled production of reactive species such as oxygen-derived ROS, nitrogen-derived reactive nitrogen species, and reactive lipid species, coupled with a weakened antioxidant defense capacity. A gradual increase in oxidative stress owing to disrupted redox regulation during aging can influence gene transcription and signal transduction pathways.

Studies have discovered chronic inflammation as a major risk factor for aging and age-related diseases [15,16]. Although inflammation as an acute response to infection and tissue damage limits harm to the host, low-grade, unresolved molecular inflammation is described as an underlying mechanism of aging and age-related diseases [17]. This chronic inflammation serves as a bridge between normal aging and age-related pathological processes. Accumulated data strongly suggest that the chronic upregulation of proinflammatory mediators (e.g., TNF-α, IL-1β, IL-6, COX-2, and iNOS) is induced during the aging process. The molecular inflammation hypothesis [16] provides insights into the interactions between age-related physiological changes and the pathogenesis of many age-related diseases [17,18].

Nuclear factor-κB plays a key role in the expression of many genes that are important in inflammatory responses [19,20]. In most cell types, inactive NF-κB complexes are sequestered in the cytoplasm via non-covalent interactions with inhibitory proteins known as IκBs. In response to multiple stimuli, including aging, ionizing radiation, endotoxins, and stress-inducing agents, the latent cytoplasmic NF-κB/IκBα complex is activated by phosphorylation of conserved serine residues in the N-terminal portion of IκBα [21,22]. This process activates NF-κB, which then translocates to the nucleus and binds to its cognate DNA-binding site in the promoter or enhancer region of specific genes. Binding sites for NF-κB are present in the promoter regions of many proinflammatory cytokines and immunoregulatory mediators, which are important in inducing acute inflammatory responses associated with critical illnesses. In particular, NF-κB plays a central role in regulating the transcription of cytokines (TNFα, IL-6, IL-1β, and IL-12), chemokines (MIP-1α and MIP-2), and other mediators involved in cardiovascular disease, liver disease, arthritis, atherosclerosis, and multi-organ system failure. Because increased activation of NF-κB leads to enhanced expression of these proinflammatory mediators, NF-κB may be a central event in the development of multiple age-related diseases [23,24].

## 3. Effects of Natural Compound-Mediated SIRT1 on Diabetes

Inflammation plays an important pathophysiological role in various disease states, including cancer, diabetes, obesity, and neurodegenerative diseases [25,26,27]. Diabetes is a metabolic disorder that causes death in the human population worldwide (mostly in the elderly) [28]. The International Diabetes Federation reported that people with diabetes show neuropathy and neuropathic pain equally [29]. The most common manifestation of diabetic neuropathy is distal symmetric polyneuropathy, which affects approximately 30% of diabetic patients with the most relevant clinical manifestations, although the incidence of distal symmetric polyneuropathy is approximately 2% per year [30]. However, its pathogenesis is unclear, and clinical and epidemiological studies indicate that oxidative stress and inflammatory processes are key pathological mechanisms in diabetic neuropathy associated with distal symmetric sensorimotor polyneuropathy.

Reactive oxygen species (ROS) are associated with the development of neuropathy in experimental diabetes. The findings in streptozotocin-injected diabetic rats suggest that oxidative stress impairs neurotransmission [31]. Overproduction and accumulation of ROS and reactive carbonyl compounds induce endoplasmic reticulum stress. Superoxide dismutase is the main antioxidant that prevents neuronal damage [32]. In diabetic neuropathy, the degeneration pathways of axonal and sensory neurons are activated, leading to distal axonal damage. The NAD^+^-dependent deacetylase SIRT1 prevents the activation of these pathways and promotes axonal regeneration [33]. A recent study demonstrated that ferroptosis, a newly identified form of regulated cell death characterized by iron-dependent dependence on ROS overproduction, leads to decreased insulin secretion [34,35]. Polysaccharides, polyphenols, and carotenoids are the major natural bioactive compounds found in *Lycium barbarum* fruit. They have many physiological and medicinal properties and have shown significant antidiabetic effects [36]. Gentiopicroside, the major active secoiridoid glycoside isolated from *Gentiana scabra* Bunge, improves diabetic glomerular fibrosis by suppressing inflammation in streptozotocin-induced diabetic mice and high glucose-induced glomerular mesangial cells [37]. However, because data on SIRT1 are scarce, little is known about its role in metabolism or its effect on diabetes.

## 4. Anti-Inflammation Effects of Green Tea in Aging

Green tea has been highlighted for its health-promoting effects in a long-term Japanese human study. They investigated mortality rates of women older than 50 who regularly participated in the Japanese tea ceremony [38,39]. Interestingly, Japanese women have markedly lower mortality rates than the general population. It was speculated that the notable effects on health and longevity of women were attributed to their continuous drinking of green tea [38,39]. Further studies revealed that the health-promoting effects of green tea are attributed to polyphenols, especially catechins, which account for more than 30% of the dry weight of green tea leaves [40]. Green tea includes various forms of catechins such as epicatechin (EC), epicatechin-3-gallate (ECG), epigallocatechin (EGC), and epigallocatechin-3-gallate (EGCG) [41]. Epigallocatechin-3-gallate accounts for approximately 65% of the total catechin content of green tea [42]. Green tea catechins are best known for their antioxidant capacity. A previous study showed that the antioxidant activity of catechins is stronger than that of vitamins C and E [43]. The antioxidant effect is attributable to the presence of phenolic hydroxyl groups in EC, ECG, EGC, and EGCG [42]. In addition to their antioxidant effects, catechins from green tea regulate various signaling pathways related to cell survival and death in neuronal and epithelial cells [44]. However, whether catechin-mediated regulation of signaling pathways has direct effects on molecular targets independent of the antioxidant effect or indirect effects by reducing oxidative stress in the cells remains unclear [45]. The chemical structure of natural compounds is shown in Table 1.

Aging is a multifactorial process in which the function of various tissues continuously decreases and susceptibility to diseases increases [46]. Based on the strong antioxidant effects of green tea, various studies have investigated whether green tea or its components affect aging-associated parameters. The effects of green tea on ethanol- and aging-induced oxidative stress were also examined in rats. Green tea administration to ethanol-intoxicated or aged rats increased the activity of antioxidant enzymes in circulation, such as superoxide dismutase and glutathione peroxidase. In addition, green tea partially recovered aging or ethanol-mediated decrease in serum antioxidants such as vitamins C, E, and A and beta-carotene [47,48]. Furthermore, green tea protected lipids and proteins against oxidative modifications induced by ethanol and aging. These data suggest that green tea is beneficial for protecting the blood against oxidative damage caused by ethanol or aging [47,48]. Furthermore, a human study showed that green tea extract consumption significantly decreased blood low-density lipoprotein oxidation, which is associated with atherosclerosis [49]. In atherosclerotic apolipoprotein E-deficient mice, supplementation with green tea extract in drinking water inhibited atherosclerosis development [50], and intraperitoneal injection of EGCG efficiently blocked atherosclerotic plaque formation [51]. In addition, green tea extract has been shown to be effective in lowering blood pressure in rats owing to its antioxidant properties [52]. Taken together, these data indicate that green tea may be helpful in preventing or treating cardiovascular diseases. Green tea components are reported to interfere with the NF-κB activity in the brain. Oral catechin supplementation in aged mice markedly decreased the protein levels of the NF-κB p65 subunit in the hippocampal formation [53]. In addition, EGCG inhibited NF-κB activation in human astrocytoma U373MG cells [54], suggesting that green tea components inhibit NF-κB activation in the brain [55]. Therefore, green tea exerts neuroprotective effects against aging-related neurodegeneration by increasing antioxidative defense and suppressing NF-κB signaling. Green tea appears to decrease aging-related tissue dysfunction in the central nervous system and peripheral tissues, at least partially by suppressing oxidative stress and interfering with inflammatory signaling.

## 5. Effect of Resveratrol through SIRT1/FoxO Signaling in Age-Related Diseases

Resveratrol (3,4,5′-trihydroxystilbene) is a plant-derived natural polyphenol with a stilbene structure. It was initially characterized as a phytoalexin as it is produced in abundance by plants undergoing various environmental stresses and exerts antimicrobial effects [56]. The most abundant dietary source of resveratrol is wine, often referred to as the French paradox [57]. It is also abundant in grape skin, raspberries, mulberries, and blueberries. Its basic structure consists of two phenolic rings bonded together and exists as two isomers (cis- and trans-), which may have different biological properties [58]. Following its discovery, multiple studies have demonstrated the beneficial effects of resveratrol in various pathologies [59,60]. It has been shown to have a wide range of biological effects, including antioxidant and anti-inflammatory effects, inhibition of lipid peroxidation, and protection from cardiovascular diseases and atherosclerosis [61,62]. Furthermore, recent studies have demonstrated its beneficial role in cancer and neurodegenerative diseases [63,64]. The antitumor activity of resveratrol has also been reported in the context of ovarian cancer both in vitro and in vivo. The proposed mechanisms include inhibition of proliferation and induction of apoptosis [65], inhibition of glucose metabolism, and combined induction of autophagy and apoptosis [66]. The effects of resveratrol on breast cancer are controversial [67] because, as a phytoestrogen, it possesses both estrogenic and antiestrogenic activities in ERɑ-positive breast cancer [68]. Many of these beneficial effects are thought to be mediated by the ability of resveratrol to reduce oxidative stress. However, its capacity to regulate the activities and expression levels of enzymes and proteins associated with cellular defense systems, inflammation, metabolism, and carcinogenesis has been reported in various studies [69,70]. Overall, the beneficial effects of resveratrol may be a result of its antioxidant properties and ability to modify intracellular signaling molecules [62].

Resveratrol has gained widespread attention owing to its ability to extend the lifespan of *Saccharomyces cerevisiae*, *Caenorhabditis elegans*, and *Drosophila* [71,72]. This effect has been shown to be mediated by the activation of SIRTs [73]. A small-molecule screen for activators led to the discovery that resveratrol can directly activate SIRT1 (mammalian SIRT), thereby increasing the lifespan of budding yeast [74,75]. Studies have revealed the roles of the SIRT family in various biological processes, such as transcriptional regulation, DNA repair mechanisms, cellular responses to stress, metabolic signaling, and aging [62,76]. These broad interactions with SIRT1 may provide various beneficial effects for resveratrol. Although it is debated whether resveratrol directly activates SIRTs, recent studies have validated the allosteric site on SIRT1 to which resveratrol binds and enhances its deacetylation activity, supporting the hypothesis that SIRT1 is a primary target of resveratrol action in vivo [77,78]. However, resveratrol has been effective chemotherapeutic and chemopreventive agent. Resveratrol are functionally pleiotropic agents, acting on multiple targets including those involved in the cell cycle, proliferation, and apoptosis [79].

Although resveratrol shows life-extension effects in lower organisms, there is little evidence supporting its life-extension effects in higher organisms. Three experiments examining the effects of resveratrol treatment on wild-type mice have been performed [41,80]. Although the experimental designs were different, resveratrol treatment did not extend the lifespan of the healthy mice. In another study using healthy rats, red wine and equivalent oral pharmacological doses of resveratrol did not extend lifespan [81]. Nevertheless, several studies have demonstrated that resveratrol treatment can increase the lifespan of metabolically compromised mammals. Baur et al. [82] first showed that resveratrol treatment prevents high-calorie diet-induced metabolic changes and reduces the risk of death owing to high-calorie intake. Furthermore, resveratrol treatment prevented the decrease in insulin sensitivity by activating AMPK/PGC1α signaling and improved mitochondrial function [83]. Subsequent studies have demonstrated that resveratrol also has anti-inflammatory and immune-modulatory effects in high-fat diet-fed mice [84]. The effects of resveratrol on metabolically compromised mice appear to involve SIRT1 and AMPK, which are directly associated with SIRT1 and exert diverse positive effects on metabolism [83]. Two studies also showed beneficial effects of resveratrol treatment on the health of rhesus monkeys fed a high-fat/high-sucrose diet [85,86]. Although the effects of resveratrol on metabolically compromised mammals are clear and reproducible, further studies are required to elucidate the exact molecular mechanisms.

Sirtuin1, which is proposed to be a central target of resveratrol in mammals, deacetylates a number of key histone and protein targets, including FoxO, NF-κB, and p53 [87,88]. Sirtuin1 has been linked to the upregulation of antioxidants, including superoxide dismutase [89]. Among several interacting non-histone proteins of SIRT1, FoxO transcriptional factors are of interest because they are closely related to lifespan in lower organisms [90]. In *C. elegans*, the extension of lifespan by Sir2 is entirely dependent on the presence of daf-16 [91], which is the only *C. elegans* ortholog of the FoxO family. Mutants in the insulin signaling pathway remain youthful and active for much longer than normal animals and can live longer; an effect that requires daf-16 activity in *C. elegans*. Numerous groups have reported an interaction between SIRT1 and the FoxO family [92,93]. Sirtuin1 mediated deacetylation of FoxO leads to decreased transactivation [92], whereas van der Horst et al. reported that SIRT1-mediated deacetylation increased the transcriptional activity of FoxO [93]. The SIRT1 agonist resveratrol acts as an antioxidant enzyme in alcohol-aflatoxin B1-induced HCC [94] and inhibits cancer cell proliferation through SIRT1-mediated modification of PI3K/AKT signaling [95]. Numerous studies have demonstrated the effects and molecular mechanisms of resveratrol on the lifespan and health of various organisms. Life extension effects are clear and reproducible in lower organisms. In mammals, resveratrol did not show life-extension effects, but improved metabolic indices in metabolically compromised models. As SIRT1 regulates metabolically important FoxO transcriptional factors in a versatile manner, and FoxO mediates the life-extension effects of resveratrol in lower organisms, FoxO might be the key player mediating the effects of resveratrol through SIRT1 activation. This study demonstrated the effects and molecular mechanisms of resveratrol in aging and cancer (Figure 1).

## 6. Effects of Natural Compounds on Liver Cancer and Other Tumors

Hepatocellular carcinoma (HCC) accounts for 70–90% of primary liver cancers, and its complex etiology and genetic polymorphisms limit the development of HCC therapy and seriously endanger human health. In recent years, clinical research on HCC has progressed; however, the mechanisms of invasion and metastasis have not yet been fully elucidated. Hepatocellular carcinoma development is closely associated with many signaling pathways, including MAPK, PI3K/Akt/mTOR, Wnt/β-catenin, and VEGF [96,97]. Aberrant lipid metabolism is a key feature in the development of malignant tumors, and obesity is highly likely to cause these tumors. Fatty acid synthase, which regulates lipid metabolism, has been demonstrated to be elevated in a variety cancer models [98].

Microtubule affinity-regulated kinase 4 (MARK4) plays an important role in energy metabolism and homeostasis. A key component of the Wnt signaling pathway, MARK4 is associated with Wnt-induced prostate cancer. MARK4 promotes adiposity and cell death by activating JNK1 and inhibiting the p38MAPK pathways. Blocking hippopotamus signaling also stimulates breast cancer cell proliferation and migration [99,100,101]. MARK4 appears to be highly relevant because it plays an important role in the survival of HCC patients [102]. Furthermore, MARK4 shares the most target proteins with gingerol, followed by resveratrol, quercetin, fisetin, apigenin, and vanillin. In contrast, mimosin shares the fewest number of target proteins [103,104]. Extracted spiropachysine A from *Pachysandra axillaris* Franch. var. stylosa (Dunn) M. Cheng inhibited HCC cell proliferation and induced methuosis through the Ras/Rac1 signaling pathways in vitro and in vivo [105]. In addition, deoxyelephantopin, a major component of *Elephantopus scaber* Linn, exerts anticancer activity against liver cancers [106] by targeting several pathways. Chlorogenic acid is a biologically active polyphenolic compound that promotes 5-fluorouracil effects in HCC cells by attenuating extracellular signal-regulated kinases (ERKs) [107] and exerts positive inhibitory effects on HCC cells (in vitro and in vivo) [108]. Chlorogenic acid also promotes oxidative stress-mediated apoptosis by activating nuclear factor erythroid 2-related factor 2 (Nrf2) in hepatocytes [109]. Genistein is a non-selective tyrosine kinase inhibitor that exhibits antitumor activity in various cancer cell types [110]. Genistein alters growth factor signaling by downregulating tyrosine kinase-regulated proteins, epidermal growth factor receptor (EGFR), and IGF-1R in transgenic adenocarcinoma in mouse prostate models [111]. In addition, genistein and curcumin have also been identified as RTK inhibitors, which cause EGFR tyrosine phosphorylation and inhibition of EGFR downstream signaling molecules Akt, ERK1/2, and STAT3 in oral squamous cell carcinoma, thus exhibiting potent cancer chemopreventive activity [112]. Animal experiments have demonstrated the preventive and therapeutic effects of natural compounds on many types of tumors, and their mechanisms have been investigated.

## 7. Ferulate from Rice Modulates NFκB Signaling

Ferulic acid (FA, 4-hydroxy-3-methoxycinnamic acid) is a natural phenolic phytochemical widely distributed in vegetables, fruits, and Chinese herbs. It was first isolated from the plant *Ferula foetida* in 1866 and is most abundant in cereal brans, where FA can reach a concentration of approximately 1351–3300 mg/100 g [113]. In plants, FA is biosynthesized by the conversion of 4-hydroxycinnamic acid to FA via caffeic acid [113,114]. Ferulic acid, like many natural phenols, is an antioxidant that quenches ROS and reactive nitrogen species by donating electrons from hydroxy and phenoxy groups [115]. The antioxidant activity of FA has been verified in several free radical-induced diseases, such as Alzheimer’s disease, cancer, cardiovascular diseases, diabetes, and skin diseases [116,117]. Ferulic acid has been receiving attention as a potent chemopreventive agent for lung, breast, and colon cancers and central nervous system tumors [118,119]. The mechanisms underlying the anticarcinogenic action of FA have been shown to block cell cycle progression, stimulate cytoprotective enzymes (superoxide dismutase and catalase), and inhibit cytotoxic enzymes and COX-2 activity in various in vitro and in vivo models [120]. In diabetes mellitus models, FA showed protective effects by regulating multiple mechanisms, such as an increase in plasma insulin levels, inhibition of blood glucose levels, expression of inflammatory cytokines, and ROS formation [116]. Combination therapy with FA and thiazolidinedione decreased most of the side effects in diabetic rats [121]. Ferulic acid supplementation for 4 weeks showed neuroprotective action by preventing an increase in the IL-1β-induced JNK pathway, leading to cell apoptosis [122].

Increased oxidative stress is a major cause of accelerated aging, and can influence gene transcription. Gene responses to oxidative stress are known to be influenced by redox-sensitive transcription factors, and the most well-known transcription factor is NF-κB. Nuclear factor-κB activation plays an important role in modulating cellular signaling mechanisms during aging and age-related diseases [123]. The balance between PTK and PTP activity is modulated by changes in the cellular redox status. Oxidative stress-induced thiol oxidation of cysteine residues affects PTK activation and PTP inactivation, leading to an eventual PTK/PTP imbalance [124]. In addition, the oxidation-dependent activation of PTK mediates many immune, growth factor, chemokine, and cytokine receptor signaling cascades through the activation of tyrosine kinases [125]. Furthermore, tyrosine kinases can activate diverse downstream serine/threonine kinases, thus stimulating transcription factors in the cytosol and the nucleus. In contrast, the oxidation of the catalytic cysteine of PTPs leads to inactivation of PP, which translates into increased activation of serine/threonine kinase cascades such as MAPKs and Akt [126]. Genistein alters growth factor signaling through downregulation of tyrosine kinase-regulated proteins, EGFR, and IGF-1R in transgenic adenocarcinoma of the mouse prostate model [111]. Recently, FA has been shown to prevent oxidative stress-induced PTK activation and PTP inactivation, and subsequent inactivation of the downstream phosphatase, PP2A, thus inhibiting renal inflammation in aged rats. Ferulic acid is currently considered to be one of the most promising dietary agents for the treatment of inflammatory diseases. The present review provides evidence that FA can inhibit the pathways of signal transduction and gene expression that play critical roles in the micro-inflammatory response during various diseases. Therefore, for anti-inflammation therapy, FA may be efficacious to inhibit oxidative stress-induced PTK/PTP-NFκB signaling. The collective evidence strongly supports the theory that the antiaging and anti-various disease effects of natural compounds are achieved via the suppression of the physiological inflammatory response, as summarized in Table 2.

## 8. Conclusions

Many diseases arise from the sites of infection, chronic irritation, and inflammation. Forkhead-box class O, SIRT1, inflammation, and metabolism play important roles in regulating aging and various diseases such as cancer and diabetes. Sirtuin1 is involved in the prevention of several diseases such as type 2 diabetes, inflammation, and cancer [128]. Studies have confirmed that activation of SIRT1 and FoxO can inhibit the NF-κB pathway and reduce inflammation. Activation of these SIRT1-dependent signaling pathways by natural compounds such as quercetin results in modulation of the levels and functions of inflammatory cytokines [129].

This concept proposes a broad perspective of the inflammatory response and creates a complex network among many inflammatory mediators that may lead to systemic chronic inflammation. Inflammation leads to inappropriate gene regulation and genomic DNA damage in various diseases. Such inappropriate gene regulation in cancer cells drives them into a proinflammatory state, resulting in altered systemic chemokine or cytokine activity. The proinflammatory environment further stresses intracellular organelles, tissues, and systems, thereby influencing the development and occurrence of various diseases. However, the secretion of proinflammatory mediators in response to internal and external stress leads to a chronic inflammatory state called inflammation. Based on observations of the effects of natural compounds, cytokines, cancer, and metabolic pathways are significantly regulated by natural compounds in various diseases (Figure 2). Taken together, a better understanding of the inflammation regulatory mechanisms is expected to provide a basis for the discovery of molecular targets that can therapeutically modulate inflammatory conditions and prevent the development of aging, diabetes, and cancer.

## Figures and Tables

**Figure 1 ijms-23-13882-f001:**
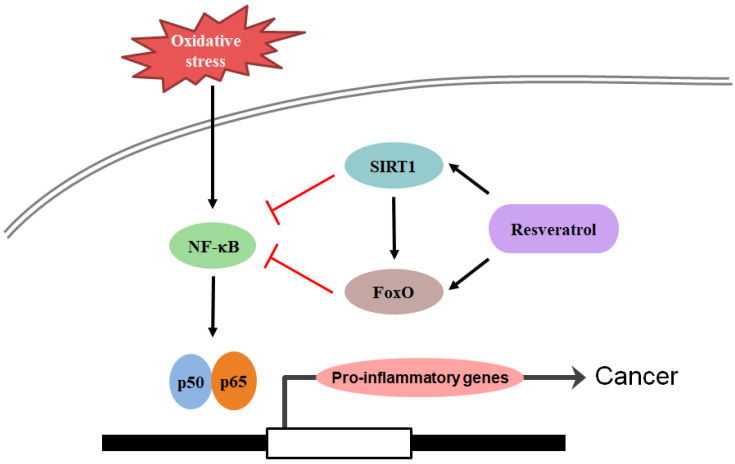
Molecular mechanism of cancer and its intervention by resveratrol. Proinflammatory molecules induced by oxidative stress causes chronic inflammation leading to cancer. Upregulation of NF-κB leads to the expression of inflammatory mediators, COX-2, iNOS, cytokines, and chemokines. All these molecules were modulated by resveratrol.

**Figure 2 ijms-23-13882-f002:**
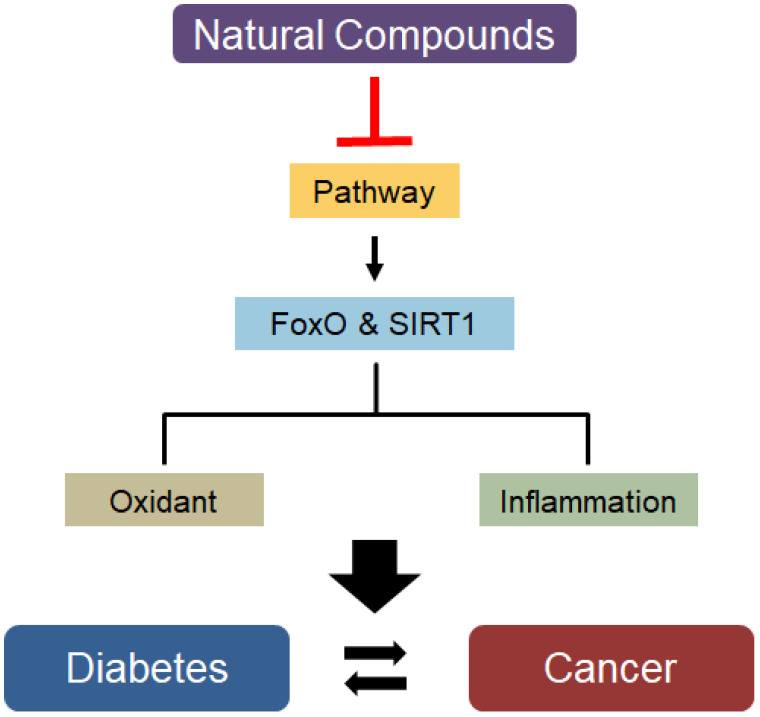
Functions of FoxO and SIRT1 targeted genes and their modifications by natural compounds and their involvements during diabetes and cancer.

**Table 1 ijms-23-13882-t001:** The chemical structure of natural compounds.

Compounds	Chemical Structures	Compounds	Chemical Structures
Carotenoids	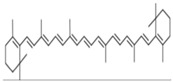	Gentiopicroside	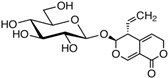
catechins	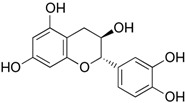	epicatechin	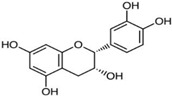
epicatechin-3-gallate	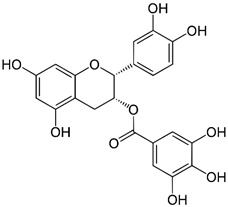	epigallocatechin	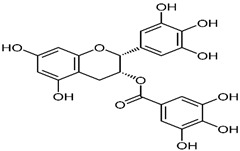
Resveratrol	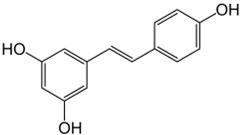	quercetin	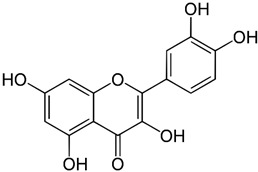
Fisetin	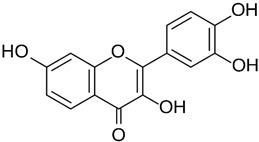	Apigenin	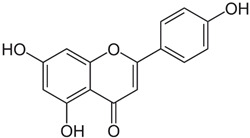
Vanillin	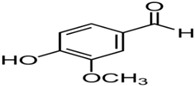	Spiropachysine A	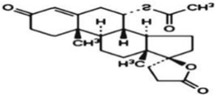
Deoxyelephantopin	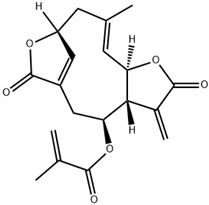	Chlorogenic acid	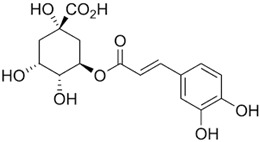
Genistein	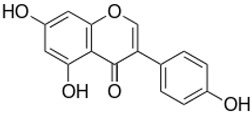	Ferulic acid	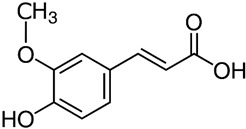

**Table 2 ijms-23-13882-t002:** Effect of natural compound in inflammation-mediated aging and various diseases.

	Factor	Aging or Diseases	Natural Compounds	References
Cytokines	TNF-α	↑	├	[51]
	IL-1β	↑	├	[69]
	IL-6	↑	├	[127]
Chemokines	IL-2	↓	├	[59]
	IL-10	↓	├	[51]
Transcription factors	NRF2	↑	├	[89,109]
	NF-κB	↑	├	[53,54]
	SIRT1	↓	├	[77,78]
	FoxO	↑	├	[91]
Signaling	MARK4	↑	├	[103,104]
	Ras/Rac1	↑	├	[105]
	ERK	↑	├	[55,107]
	Akt	↑	├	[85,112]
	AMPK	↓	├	[65]

├, reversed compare with aging or diseases.

## Data Availability

The data presented in this study are available on request from the corresponding author. The data are not publicly available due to privacy.

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
