# Peer review of "Transcription Factors as Targets of Natural Compounds in Age-Related Diseases and Cancer: Potential Therapeutic Applications"

_ijms, 2022, doi:10.3390/ijms232213882_

Round 1
Reviewer 1 Report
In this manuscript, the authors described the use of natural products in age-related diseases and cancer via modulation of sirtuins and in particular SIRT1.
Overall, this manuscript comprehensively summarizes biological and medicinal aspects of the use of natural compounds in this specific setting and would highly be expected to make concrete contribution to the field.
I would recommend acceptance of this manuscript following correction of few minor points listed below:
1) The authors should provide a more critical opinion on the actual modulation of natural products of sirtuins since they have pleiotropic effects. Polyphenol-based drugs have multiple targets and may interfere with biological assays and are indeed classified as PAINs (Pan-assay interference compounds). For instance, resveratrol has been indicated to activate SIRT1 but also modulate many other targets. The authors should mention that.
2) Chemical structures of the mentioned compounds should be shown.
3) In introducing natural products as SIRT modulators, for completeness, the authors should cite the following review that nicely summarize sirtuine modulators and goes in depth into the mode of action of resveratrol and synthetic analogues: https://doi.org/10.4155/fmc-2022-0031
Author Response
Reviewer I
In this manuscript, the authors described the use of natural products in age-related diseases and cancer via modulation of sirtuins and in particular SIRT1.
Overall, this manuscript comprehensively summarizes biological and medicinal aspects of the use of natural compounds in this specific setting and would highly be expected to make concrete contribution to the field.
I would recommend acceptance of this manuscript following correction of few minor points listed below:
Response: Thanks for your advice. We have carefully corrected and improved the quality of manuscript as many as possible.
1) The authors should provide a more critical opinion on the actual modulation of natural products of sirtuins since they have pleiotropic effects. Polyphenol-based drugs have multiple targets and may interfere with biological assays and are indeed classified as PAINs (Pan-assay interference compounds). For instance, resveratrol has been indicated to activate SIRT1 but also modulate many other targets. The authors should mention that.
Response: As suggested, we clarified this point in the revised MS (line 230) and add reference.
2) Chemical structures of the mentioned compounds should be shown.
Response: Thanks for your comments. In accordance to your suggestion, we added the statement of table 1 in the revised manuscript
3) In introducing natural products as SIRT modulators, for completeness, the authors should cite the following review that nicely summarize sirtuine modulators and goes in depth into the mode of action of resveratrol and synthetic analogues: https://doi.org/10.4155/fmc-2022-0031
Response: Thanks for suggestion. We modified the sentence with proper citations (line 64).

Reviewer 2 Report
The main problem this reviewer has with the review article “Therapeutic application of natural compounds in age-related diseases and cancer: Transcription factors as a potential mechanism” by Kim et al. is the scarce organization of the MS and a little bit of confusion.
This reviewer wishes to provide some suggestions to improve this MS, as follows.
1. English must be carefully edited, especially regarding the organization of the sentences.
2. This reviewer feels the title’s not appropriate. Indeed, the authors stand at the end of the title “transcription factors as a potential mechanism”, but mechanism of what? They mean mechanism of cells’ response? This reviewer feels that TFs are rather targets of natural compounds than mechanisms (e.g. resveratrol activates SIRT1).
Maybe a more reasonable title should be “Transcription factors as targets of natural compounds in age-related diseases and cancer: potential therapeutic applications”.
This is only a suggestion; however, the authors have to explain in the title to which mechanism they refer to.
3. Abstract: I believe that the authors must add NF-kB as a “target” of this review in the abstract and introduction section. Indeed, a large part of the review is also dedicated to NF-kB (actually, a major part).
3. The Introduction section must be better organized. It seems a series of phrases stuck without any logical consequence.
4. Page 1, line 32: please, mention the cells repsonsible for the secretion of the cytokines and chemokines mentioned.
5. Page 2, line 48: eliminate the word “However”.
6. Page 2 and following: this reviewer would better organize section 3 of the Review article in this way:
a. A descriptive paragraph on Sirtuins and the role of Sirt1 in oxidative stress, inflammation and disease
b. Descriptive paragraphs on NF-kB and FoXO (this latter indicated by the authors in the abstract as the major TF they will discuss in the paper), both targets of Sirt1.
7. Move paragraph 6 after new paragraph 3, as it depicts the effects of herbal compounds on NF-kB
8. Move paragraph 7 after new paragraph 4, as, again, NF-kB is mentioned
9. I would move section 3 at the end, before the Conclusions section. Furthermore, I will enlarge this section, including other tumours than the sole HCC, such as lymphomas, breast cancer.
Author Response
Reviewer II
The main problem this reviewer has with the review article “Therapeutic application of natural compounds in age-related diseases and cancer: Transcription factors as a potential mechanism” by Kim et al. is the scarce organization of the MS and a little bit of confusion.
This reviewer wishes to provide some suggestions to improve this MS, as follows.
Response: Thanks for your advice. We have carefully corrected and improved the quality of manuscript as many as possible.
- English must be carefully edited, especially regarding the organization of the sentences.
Response: Thanks for your comment. We were more careful in writing English sentences. Grammar and English sentences have been corrected by a professional proofreading company. http://www.editage.co.kr
- This reviewer feels the title’s not appropriate. Indeed, the authors stand at the end of the title “transcription factors as a potential mechanism”, but mechanism of what? They mean mechanism of cells’ response? This reviewer feels that TFs are rather targets of natural compounds than mechanisms (e.g. resveratrol activates SIRT1).
Maybe a more reasonable title should be “Transcription factors as targets of natural compounds in age-related diseases and cancer: potential therapeutic applications”.
This is only a suggestion; however, the authors have to explain in the title to which mechanism they refer to.
Response: Thanks for your comments. We changed title.
- Abstract: I believe that the authors must add NF-kB as a “target” of this review in the abstract and introduction section. Indeed, a large part of the review is also dedicated to NF-kB (actually, a major part).
Response: Thanks for your comment. We added clear description in the revised abstract.
- The Introduction section must be better organized. It seems a series of phrases stuck without any logical consequence.
Response: Thanks for your comment. We modified it in introduction section
- Page 1, line 32: please, mention the cells repsonsible for the secretion of the cytokines and chemokines mentioned.
Response: We add to revised MS (line 33).
- Page 2, line 48: eliminate the word “However”.
Response: Thanks for your comment. We delete.
- Page 2 and following: this reviewer would better organize section 3 of the Review article in this way:
- A descriptive paragraph on Sirtuins and the role of Sirt1 in oxidative stress, inflammation and disease
- Descriptive paragraphs on NF-kB and FoXO (this latter indicated by the authors in the abstract as the major TF they will discuss in the paper), both targets of Sirt1.
Response: Thanks for your comment. In accordance to your suggestion, we arranged.
- Move paragraph 6 after new paragraph 3, as it depicts the effects of herbal compounds on NF-kB
Response: Thanks for your comment. In accordance to your suggestion, we moved.
- Move paragraph 7 after new paragraph 4, as, again, NF-kB is mentioned
Response: Thanks for your comment. In accordance to your suggestion, we moved.
- I would move section 3 at the end, before the Conclusions section. Furthermore, I will enlarge this section, including other tumours than the sole HCC, such as lymphomas, breast cancer.
Response: Thanks for your comment. In accordance to your suggestion, we moved.

Round 2
Reviewer 2 Report
This reviewer feels the work by Kim et al., improved. Before publication, two minor points have to be amended. Line 33: this reviewer suppose the authors are talking of "macrophages" not "microphages" (although microphages have been also reported). Please, add "s" both to monocyte and macrophage.
Paragraph 6: this reviewer would change the title in "Effects of natural compounds on liver cancer and other tumors".
Author Response
Comment
This reviewer feels the work by Kim et al., improved. Before publication, two minor points have to be amended.
Line 33: this reviewer suppose the authors are talking of "macrophages" not "microphages" (although microphages have been also reported). Please, add "s" both to monocyte and macrophage.
Response: Thanks for your comments. We add both macrophages and microphages, and add "s" both to monocyte and macrophage (line 33-34).
Paragraph 6: this reviewer would change the title in "Effects of natural compounds on liver cancer and other tumors".
Response: Thanks for your advice. we change title of paragraph 6 (line 280).